# Clinical Impact and Safety of Non-Target Punctures (NTP) during Portal Vein Access in TIPS Procedure

**DOI:** 10.3390/biomedicines11061630

**Published:** 2023-06-03

**Authors:** Sasikorn Feinggumloon, Zachary Haber, Sammy Saab, Fady Kaldas, Navid Eghbalieh, Thanh T. Luong, Justin P. McWilliams, Edward Wolfgang Lee

**Affiliations:** 1Division of Interventional Radiology, Department of Radiology, UCLA Medical Center, David Geffen School of Medicine at UCLA, Los Angeles, CA 90095, USA; sasikorn.fei@gmail.com (S.F.); zhaber@mednet.ucla.edu (Z.H.); navide123@yahoo.com (N.E.); ttluong@health.ucsd.edu (T.T.L.); jumcwilliams@mednet.ucla.edu (J.P.M.); 2Division of Hepatology, Department of Medicine, UCLA Medical Center, David Geffen School of Medicine at UCLA, Los Angeles, CA 90095, USA; ssaab@mednet.ucla.edu; 3Division of Liver and Pancreas Transplantation Surgery, Department of Surgery, UCLA Medical Center, David Geffen School of Medicine at UCLA, Los Angeles, CA 90095, USA; fkaldas@mednet.ucla.edu

**Keywords:** transjugular intrahepatic portosystemic shunt (TIPS), complications, non-target puncture, cirrhosis, portal vein access, angiography, portal vein, portal hypertension, variceal bleeding, refractory ascites

## Abstract

Background: Although non-target puncture (NPT)-related complications are well known to clinicians performing TIPS, there is no NTP-focused study to assess the true clinical sequalae of NTP-related complications. In this study, the aim was to evaluate the incidence, safety, clinical outcomes and complications related to NTPs during the portal access of TIPS procedures. Methods: A retrospective review of 369 TIPS procedures from October 2007 to September 2019 was performed. We identified inadvertent NTPs, including biliary, hepatic artery, lymphatic and capsular punctures. Next, the medical records and images were reviewed and analyzed to assess the safety and clinical outcomes of these cohorts. Results: A total of 71 NTPs were identified in 56 patients (15.18% of 369 patients). Of 369 TIPS patients, there were (1) 28 biliary punctures (7.6%), (2) 16 extracapsular punctures (4.3%), (3) 15 lymphatic punctures (4.1%) and (4) 12 hepatic artery punctures (3.3%). The overall complication rate was 2.2% (8/369). Based on the Clavien–Dindo classification, three patients (0.8%) had a minor complication. In addition, five patients (1.4%) experienced grade II–V major complications, such as symptomatic hemoperitoneum, arterio-biliary fistula or hemorrhagic shock leading to death. Mortality (0.5%) was only caused by extracapsular puncture combined with other NTP. Conclusions: NTPs during the portal access of TIPS procedures are associated with low complication risk. However, when extracapsular punctures are combined with other NTPs, a more severe complication, including mortality, can occur. Nevertheless, all patients with NTP should be closely monitored at a higher level of care after TIPS placement.

## 1. Introduction

A transjugular intrahepatic portosystemic shunt (TIPS) is a widely accepted procedure to treat the complications of portal hypertension. It relieves portal hypertension by creating a shunt between portal and systemic circulations. Two most common indications are for the secondary treatment of variceal bleeding and refractory ascites [1]. Although TIPS is considered minimally invasive and shows high clinical efficacy, it may cause complications, such as intraperitoneal bleeding, hemobilia, TIPS dysfunction, hepatic encephalopathy and liver failure. Some of these complications can be resolved and improved with technical advancements, such as use of ePTFE stent graft to improve TIPS patency. However, some complications continue to occur (e.g., non-target puncture (NTP) during portal access, causing intraperitoneal bleeding) and are reported persistently despite these technical improvements [1,2,3,4].

NTP is considered one of the most serious complications of the TIPS procedure, occurring in 0.6–4.3% of the patients undergoing TIPS [3,5]. It is caused by inadvertent needle puncture through non-target structures, such as the hepatic artery, bile duct, intrahepatic lymphatic or liver capsule. Due to the technical challenges of performing intrahepatic access of the portal vein, many image-guided techniques were previously proposed to help identify the portal vein, such as direct portography [6,7], wedge hepatic venography, balloon occlusion venography [8,9,10] or imaging guidance [11,12,13,14,15]. However, these image-guided techniques caused additional complications [16,17,18,19,20]. Furthermore, many of the techniques did not improve clinical outcomes of patients undergoing the TIPS procedure or reduce complications from NTP [11,21,22].

Although NTP-related complications are well known to clinicians performing TIPS, there is no NTP-focused study that assesses the true clinical sequalae of NTP-related complications. Therefore, in this study, we aimed to evaluate the incidence, safety and clinical outcomes of complications caused by NTPs during TIPS procedures.

## 2. Materials and Methods

After obtaining approval from our Institutional Review Board (IRB) (IRB# 10-000464), 369 angiograms and medical records were retrospectively reviewed, and clinical data were collected from patients who underwent TIPS from December 2007 to September 2019 at a single institution. TIPS procedures were performed by interventional radiologists with varying degrees of experience, ranging from 1 year to 35 years.

### 2.1. TIPS Procedure

The standard TIPS technique was used [23,24]. In brief, with the right internal jugular venous access, the Rosch–Uchida transjugular access set (Cook Medical, Bloomington, IN, USA) was placed into the hepatic vein. After the catheter was placed into a desirable location in the hepatic vein, the needle was advanced intra-hepatically through parenchyma into the portal vein. This step was where the NTP of other structures could have occurred. Once the needle access of the portal vein was confirmed, the portal pressure and systemic pressure in the right atrium were measured to calculate the portosystemic pressure gradient (PSPG). A porto-venogram was performed to evaluate the anatomy and flow dynamics. Next, the portal vein was accessed further with a sheath, and the Viatorr endoprosthesis stent-graft (Gore Medical Inc., Flagstaff, AZ, USA) was placed through the sheath, connecting the portal vein to the HV/IVC confluent. An appropriately sized balloon (6–10 mm) was then used to dilate the stent-graft. Through the post-stented and ballooned TIPS, post-TIPS PSPG was measured, and a final porto-venogram was obtained. Variceal embolization was performed in selected cases.

### 2.2. Data Collection and Statistical Analysis

Each angiographic image of the TIPS procedures was collected and evaluated for inadvertent non-target puncture of other structures. Demographic data, MELD scores, Child–Pugh scores, times from procedure to discharge, readmission rates, liver transplant rates, outcome parameters, relevant clinical data, and pre- and post-imaging analyses were collected. Continuous data were summarized as means with standard deviation or median with inter-quartile ranges, depending on the distribution of the data. Qualitative variables, including gender, TIPS indication, emergent case, hemodynamic success, readmission rate and Child–Pugh score, were shown as raw numbers and percentages. Quantitative variables, including age, MELD score, creatinine, INR, total bilirubin, sodium, albumin and fluoroscopic times and length of hospitalization, are reported as means with standard deviation (SD). The Student’s *t*-test was used to compare the differences between continuous variables, and either Pearson’s chi-square test or Fisher’s exact test was used to compare categorical variables between the groups. *P* values < 0.05 were regarded as statistically significant. All statistical analyses were performed using SPSS software version 22.0 (IBM, Chicago, IL, USA).

## 3. Results

All of the 369 angiographic TIPS images were retrospectively reviewed for inadvertent NTP of vital structures, including bile ducts, hepatic artery, intrahepatic lymphatic and capsular punctures (Figure 1). All NTP data were compared to the entire TIPS cohort (Table 1). A total of 71 non-target punctures (with 11 combinations of different NTPs, Figure 2) were identified among 56 patients (15.2% of all patients). Among 56 patients, 39 were male and 17 were female. Of 56 patients, 51 (91.2%) achieved hemodynamic success. A total of 13 patients (23.2%) underwent emergent TIPS. The number of each type of NTPs (Table 2) were 28 biliary punctures (7.6%), 16 extra-capsular punctures (4.3%), 15 lymphatic punctures (4.1%) and 12 hepatic artery punctures (3.3%). The mean (SD) length of stay of the patients was 10.2 ± 16.8 days, which was not statistically different to non-NTP group (12.34 ± 42.5, *p* = 0.498). The readmission rate within one month of discharge was 48.2%, and the liver transplant rate was 21.4%. The average Pre-TIPS MELD-Na score was 17.6 ± 9.1. Child–Pugh scores were A (4, 7.1%), B (28, 50.0%) and C (21, 37.5%). Three of the patients were not able to be scored due to missing data. Notably, in comparison, one-month readmission rates for NTP patients were statistically higher than for the entire TIPS cohorts, i.e., 48.2% vs. 18.2% (*p* < 0.0001). Otherwise, all clinical parameters were not significantly different.

Complications were observed in eight patients (2.2% of all TIPS patients, or 14.3% of NTP patients), all of whom were graded according to the Clavien–Dindo classification (Table 3) [25]. Grade I complications were found in three cases, namely one case of focal segmental biliary ductal dilatation, one case of intra-operative mild hypotension and one case of transient elevation of AST/ALT. These three cases did not require any additional treatment. Grade II complications were found in two patients who had cases of hemoperitoneum that required blood transfusion. No Grade III complications were noted. A Grade IVB complication was found in one patient who experienced hemorrhagic shock following TIPS procedure, which led to multi-organ failure and death within 2 months of TIPS insertion. Grade V complications were found in two cases. One case was caused by intraperitoneal bleeding and hypovolemic shock, which led to multi-organ failure and death within a week. In the other case, the patient suffered from bleeding and hypovolemic shock immediately after TIPS placement. The patient was immediately brought back to IR, and the hepatic angiogram was performed, which demonstrated an arterio-biliary fistula that was successfully embolized. However, the patient did not recover from multi-organ failure and died about 1 month later.

## 4. Discussion

Multiple studies showed that non-target puncture (NTP) is difficult to avoid in the TIPS procedure, as the artery, lymphatics and bile ducts are in close proximity to the portal vein [26,27,28,29,30]. Uflacker et al. published a pathological study demonstrating a co-existence of the portal vein, bile ducts or hepatic artery along the path of portal puncture, extending from RHV to portal vein bifurcation. Uflacker noted that 96% of hepatic arteries and bile ducts from segments VII and VIII are above portal vein bifurcation when the right portal vein is targeted as an access [31]. A small cirrhotic liver causes these structures to be cramped in a confined space and forced into closer proximity. Therefore, TIPS in a small liver may cause more NTPs, as well as biliary and vascular injury, compared to TIPS in a liver of normal size. Moreover, in a microscopic view of the portal triads, the portal veins were surrounded by arterial branches, bile ducts and lymphatics. Therefore, NTP is inevitable and common during TIPS procedures. In addition, biliary and hepatic arterial anatomic variants are very common, occurring in up to 45% and 42% of patients, respectively [32]. These anatomic variants can also contribute to NTP during TIPS. Pathophysiologically, a portal vein puncture could be more challenging in a cirrhotic liver due to portal vein thrombosis or narrowing of the portal system, which may also increase the chance of NTP.

Several studies reported NTP-induced complications during TIPS procedures and identified capsular puncture as a cause of morbidity and mortality from intraperitoneal bleeding [5,17,33,34]. Freedman et al. found capsular puncture in 30% of all TIPS cases, though only one case was found to cause a serious hemoperitoneum. Moreover, Loffroy et al. found that capsular puncture may occur in up to 33% of cases, and 1–2% of those transcapsular punctures result in intraperitoneal hemorrhage. In addition, Haskal et al. reported a case of hepatic artery injury from a TIPS procedure [35]. Although arterial injury can lead to a lethal complication, they concluded that arterial injury is uncommon. Several studies reported fistulous connections between the bile ducts and hepatic artery, as well as between the portal vein and hepatic vein. For example, Willner et al. reported a case in which a patient had recurrent infection after TIPS placement and found a porto-biliary fistula in the explant after liver transplantation [36]. In another case, Menzel et al. found a hepatic arterio-biliary fistula that caused a massive hemobilia post-TIPS [37]. Lastly, Mallery et al. found a hepatic veno-biliary fistula in a patient who developed persistent sepsis after the TIPS procedure [27]. However, none of these reports or other studies conducted a comprehensive analysis of complications and outcomes of NTP in TIPS patients.

David et al. assessed peri-procedural complications caused by NTP in trans-abdominal ultrasound-guided portal vein access in TIPS [22]. This study suggested that an ultrasound-guided portal vein puncture may be a safer method, with a rate of 5.4% for overall puncture-related complications in 224 TIPS. The study found a lower rate of complication compared to prior studies [26,34]. However, this study found a higher complication rate than we found, which was 2.2%. In addition, this study failed to describe NTP without complications and did not compare its outcomes to any control group (non-US guided TIPS). Therefore, it is not necessarily convincing that US-guided TIPS is safer.

The 1-month mortality rate due to NTP during the TIPS procedure was 0.5% (2/369). None of the NTP patients experienced biliary tract infection, biliary obstruction, hepatic artery aneurysm or lymphatic leakage within 30 days of TIPS placement. The mortality rate is relatively low compared to the larger rate found by Barton et al., which was a 1.7% procedural mortality rate from hemoperitoneum, retroperitoneum, mediastinum, laceration of hepatic artery, portal vein, liver capsule and right heart failure [38].

Our study systematically collected data from all NTP cases that occurred during TIPS procedures to identify the types of punctures that lead to morbidity or mortality. We found NTP-related complications in eight patients. Notably, four out of five patients with clinically significant complications had a combination of capsular puncture and NTP of other structures. Accordingly, all mortalities were associated with a combination of capsular puncture and other NTP. The remaining patient with a clinically significant complication from NTP had markedly decreased hemoglobin levels (dropping from 8 to 5.9), thus requiring transfusion. This patient had hepatic artery NTP without capsular puncture or evidence of intraperitoneal hemorrhage. Therefore, the drop in hemoglobin may be contributed to other causes, such as hemodilution or hemolysis. After the transfusion, the patient was stabilized without further intervention/treatment. None of the inadvertent isolated capsular punctures led to clinically significant symptoms. There is no statistically significant difference in NTP complications, post-TIPS complications, readmission rates, lengths of stays, or mortality rates in either the combination of other non-capsular puncture NTPs or the isolated capsular puncture cohort. Similarly, no clinically significant outcomes were solely caused by arteries, lymphatics, bile ducts or combined multiple NTPs without capsular puncture, including NTP complications, post-TIPS complications, readmission rates and lengths of stays. None of the NTP patients in these groups died. Therefore, we recommend even closer clinical observation and, if necessary, additional imaging in patients who had an inadvertent capsular puncture combined with other NTPs. This approach may help clinicians to avoid delayed diagnosis of complications, such as peritoneal hemorrhage, as well as minimize prolonged hypovolemic shock from bleeding.

Our study has a number of limitations that warrant further discussion. Firstly, it was a retrospective single-center study. There was a selection bias for patient inclusion, given that patients were only included if they had imaging evidence of NTP during their TIPS procedure. Therefore, some patients who had NTPs without images could have been excluded. Future prospective studies are needed to eliminate this selection bias. Another limitation is that the follow-up data are based on the medical records within our system, and it may or may not include hospitalizations at outside facilities. Given that those outside hospital records were not available, some data may be missing. Again, a larger prospective study may address this issue and potentially eliminate this limitation. Lastly, although data are from a single-center experience, several interventional radiologists performed TIPS at our institution with a range of experiences and skill sets. This fact may have affected the outcomes of this study. However, in evaluation, it was noted that the NTP-related complications were not associated with the length of experience or skill sets.

## 5. Conclusions

In conclusion, non-target puncture injury during TIPS is not uncommon. An isolated NTP of any major intrahepatic structures, including arteries, bile ducts, lymphatics or liver capsule, does not appear to cause any clinically significant complications. However, morbidity and mortality increase if a NTP of the liver capsule is combined with other non-target punctures. Therefore, closer observation and a higher level of monitoring, including additional imaging, may be warranted to prevent unexpected clinical outcomes.

## Figures and Tables

**Figure 1 biomedicines-11-01630-f001:**
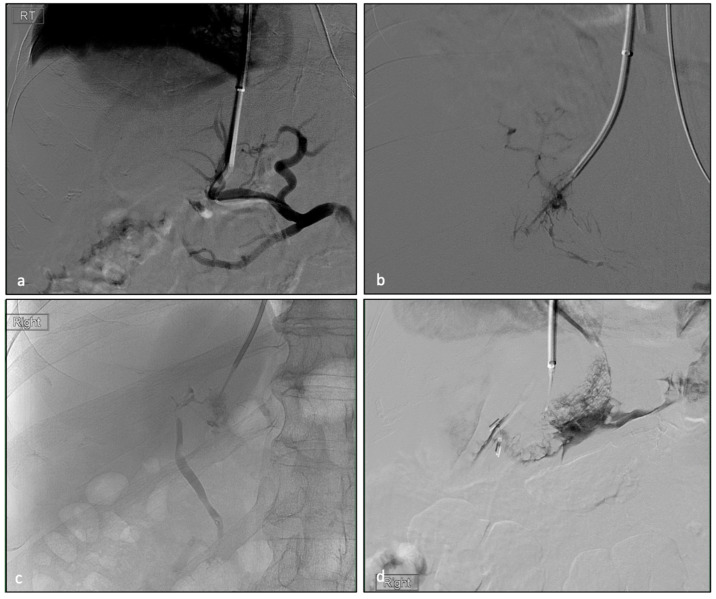
Angiograms during portal access of TIPS procedures show inadvertent non-target puncture (NTP) of (**a**) hepatic artery, (**b**) lymphatic system, (**c**) biliary system and (**d**) capsular puncture.

**Figure 2 biomedicines-11-01630-f002:**
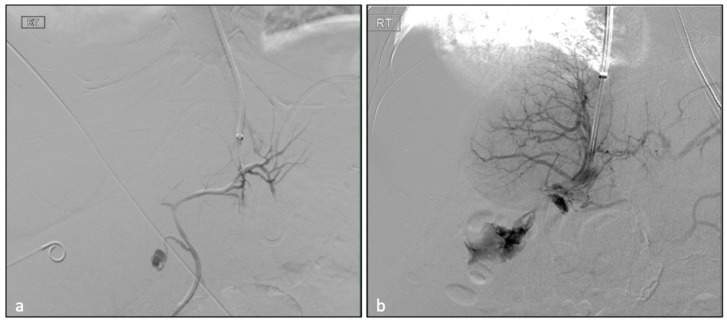
Angiograms during portal access of the TIPS procedures show combined inadvertent puncture of (**a**) hepatic artery and biliary system and (**b**) hepatic artery and capsular puncture in a single patient.

**Table 1 biomedicines-11-01630-t001:** Patient characteristics.

Parameter	All TIPS Patients (*n* = 369)	NTP Patients (*n* = 56)	*p*
Gender			
Male	241 (65.3%)	39 (69.6%)	
Female	128 (34.7%)	17 (30.4%)	
Age (years)	56.2 ± 15.1	57.95 ± 13.5	0.377
Emergency			
Yes	77 (20.9%)	13 (23.2%)	0.689
Indication			
Bleeding	127 (34.4%)	22 (39.3%)	
Refractory ascites	234 (63.4%)	34 (60.7%)	
Pre-TIPS			
Total bilirubin	3.4 ± 7.2	3.5 ± 5.4	0.969
INR	1.4 ± 0.5	1.4 ± 0.4	0.900
Creatinine	1.4 ± 1.1	1.5 ± 1.4	0.578
Sodium	134.3 ± 11.2	134.4 ± 5.6	0.895
Albumin	3.1 ± 0.7	2.9 ± 0.5	0.001
Pre-TIPS MELD-Na score	17.9 ± 11.5	17.6 ± 9.1	0.808
Child–Pugh class			
A	29 (7.9%)	4 (7.1%)	
B	226 (61.2%)	28 (50.0%)	
C	102 (27.6%)	21 (37.5%)	
Hemodynamic success	338 (91.6%)	51 (91.1%)	0.801
LOS (days)	12.34 ± 42.5	10.2 ± 16.8	0.498
Readmission rate (within 1 month)	67 (18.2%)	27 (48.2%)	0.000
Liver transplant rate	54 (14.6%)	12 (21.4%)	0.191

**Table 2 biomedicines-11-01630-t002:** Number of non-target punctures (NTPs) in overall TIPS.

NTP Type	Number of NTPs	% of Each NTP in Total NTPs
Biliary punctures	28 (7.6%)	34%
Extra-capsular punctures	16 (4.3%)	20%
Lymphatic punctures	15 (4.1%)	18%
Hepatic artery punctures	12 (3.3%)	15%
Combination of any NTP	11 (3.0%)	13%

**Table 3 biomedicines-11-01630-t003:** Summary of complications related to NTP according to Clavien–Dindo classification.

ID	Biliary Puncture	Hepatic Artery Puncture	Lymphatic Puncture	Capsular Puncture	Clavien-Dindo Grading Complication in 30 Days	Complications	LOS (Days)
6	X	0	0	0	1	Focal segment 7 biliary ductal dilatation	6
8	X	0	0	X	4B	Multiorgan failure	56
30	0	X	0	0	2	Hemoperitoneum need, blood transfuse	13
35	0	X	0	X	2	Hemoperitoneum need, blood transfuse	2
41	0	0	X	0	1	Mild hypotension	2
43	0	0	X	X	5	Hemoperitoneum, hypovolemic shock	8
51	X	X	X	X	5	Arteriobiliary fistula	26
54	0	0	0	X	1	Mild hypotension	4

X denotes that specific NTP; 0 denotes for no NTP.

## Data Availability

The data presented in this study are available on request from the corresponding author. The data are not publicly available due to privacy restrictions and HIPAA compliance issues.

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
