# Peer review of "Clinical Impact and Safety of Non-Target Punctures (NTP) during Portal Vein Access in TIPS Procedure"

_biomedicines, 2023, doi:10.3390/biomedicines11061630_

Round 1
Reviewer 1 Report
Thanks for the chance to review this retrospective cohort study of TIPS looking at NTP and resultant complications. The study is largely clear and well performed (with good discussion of its limitations), although the result might benefit from being more clearly presented in a few areas.
1) Table 3: please provide a key to explain what X and O mean.
2) Consider reporting NTP proportions within NTP rather than as a fraction of the overall cohort, as readers are likely to be interested in the makeup of NTPs, which is hard to garner from the results as presented. eg lines 104-106
Thanks for the chance to review this study.
Author Response
Dear Editors and Reviewers,
We have made all the changes as suggested.
1) Table 3: please provide a key to explain what X and O mean.
- The keys for X and O is provided at the bottom of the table
2) Consider reporting NTP proportions within NTP rather than as a fraction of the overall cohort, as readers are likely to be interested in the makeup of NTPs, which is hard to garner from the results as presented. eg lines 104-106
- We feel that both information are important so we included the makeup of NTPs in the table 2 to provide both information.
THANK YOU
Reviewer 2 Report
The authors analyze a topic which is of interest – accidents and complications in TIPS.
The presentation is clear, comprehensive and well documented.
The references are appropriate, up-to-date and contain 38 titles.
I found no self-citations.
The figures (6 angiograms) are appropriate and mandatory for sustaining the topic.
The 3 tables offer concentrated information on the topic.
I found no plagiarism.
The discussions and conclusions are coherent and connected to the content.
In my opinion the paper fits the journal and the language is correct and understandable.
I recommend the paper to be accepted.
Author Response
Dear Editors and Reviewer,
Thank you so much for your comments.
no changes are maded.